# Metasurfaces for Amplitude-Tunable Superposition of Plasmonic Orbital Angular Momentum States

**DOI:** 10.3390/ma15186334

**Published:** 2022-09-13

**Authors:** Yuqin Zhang, Jianshan An, Xingqi An, Xiangyu Zeng, Changwei He, Guiyuan Liu, Chuanfu Cheng, Hongsheng Song

**Affiliations:** 1School of Science, Shandong Jianzhu University, Jinan 250101, China; 2School of Physics and Electronics, Shandong Normal University, Jinan 250014, China

**Keywords:** metasurface, orbital angular momentum, surface plasmon polaritons

## Abstract

The superposition of orbital angular momentum (OAM) in a surface plasmon polariton (SPP) field has attracted much attention in recent years for its potential applications in classical physics problems and quantum communications. The flexible adjustment of the amplitudes of two OAM states can provide more freedom for the manipulation of superposed states. Here, we propose a type of plasmonic metasurface consisting of segmented spiral-shaped nanoslits that not only can generate the superposition of two OAM states with arbitrary topological charges (TCs), but also can independently modulate their relative amplitudes in a flexible manner. The TCs of two OAM states can be simultaneously modulated by incident light, the rotation rate of the nanoslits, and the geometric parameters of the segmented spiral. The relative amplitudes of the two OAM states are freely controllable by meticulously tuning the width of the nanoslits. Under a circularly polarized beam illumination, two OAM states of opposite TCs can be superposed with various weightings. Furthermore, hybrid superposition with different TCs is also demonstrated. The presented design scheme offers an opportunity to develop practical plasmonic devices and on-chip applications.

## 1. Introduction

The angular momentum of a paraxial beam can be classified as spin angular momentum (SAM) and orbital angular momentum (OAM) [1]. SAM manifests as the handedness of circular polarizations of light [2], while the OAM is carried by a vortex beam with a helical phase wavefront, described by exp (*ilθ*), where *l* is the topological charge (TC) of the vortex, and *θ* is the azimuthal angle, representing the *lħ* value of the OAM per photon [3]. Due to their helical characteristics, vortex beams have been widely applied to metrology [4], optical micromanipulation [5], and optical communications [6,7]. The unlimited TCs and inherent orthogonality also create the possibility of exploiting OAM for high-capacity optical information technologies [8,9] and the emerging field of quantum information science [10,11]. More interestingly, the superposition of two or more OAM states with different TCs can produce structured fields with complex phase distributions [12,13]. An equal-weighted linear superposition of OAM states with opposite signs can produce a vector beam with a petal-like intensity distribution, which has invoked wide investigations and important applications of tight focusing and laser processing [14,15]. The superposition of high-order OAM states presents a photonics gear-like structure beam for ultrasensitive angular measurement and spin object detection [16,17,18]. Multi-OAM states can also be used to generate arbitrary superposed states, which have been exploited to gain deep insight into quantum correlations, better understand the violation of Bell-type inequalities, and observe entangled vortex links [19,20]. In addition, a variety of relative amplitudes of two vortex beams are involved in superposition and can serve as a carrier of information in wireless communications [21]. Various tools have been established to produce superposed OAM models, such as *q* plates [22], cylindrical lenses [23], spiral phase plates [24], computer-generated holograms [16], spatial light modulators [25], and inhomogeneous birefringent elements [26]. Recently, multifunctional metasurfaces have provided an alternative strategy for manipulating the superposition of vortex beams, leading to the development of miniaturization and multifunctional metasurface devices. In 2017, Yue et al. proposed a reflective plasmonic metasurface to realize the superposition of OAM states in four channels [27]. By designing a dielectric metasurface for simultaneous control of the amplitude and phase, Li et al. achieved polarization multiplexed OAM superposed states [28]. In general, most superposed OAM states generated via metasurface elements are performed in the far field.

Surface plasmon polaritons (SPPs) that propagate along the metal–dielectric interfaces have attracted considerable attention due to their ability to carry surface-confined OAMs and form near-field OAM states [29,30,31]. Various structures, including circular slits [32,33,34], chiral geometric structures [35], and well-arranged units [36], have been used to construct a plasmonic OAM. Moreover, chiral geometric structures can be combined with rotated slits to provide flexible control over the topology of a plasmonic vortex [37] and began to involve the superposition of the plasmonic OAM state. Zang et al. designed double-ring distributed slit arrays for investigating the manipulation of terahertz near-field OAMs and the corresponding superposed states [38]. In our previous work, we proposed the plasmonic metasurfaces with a single-slit of the identical dimension as the unit to generate an equal-weighted superposition of two plasmonic OAM states [39], while their incapability of amplitude modulation limited the generation of more complicated SPP fields. Recently, we made use of the nanoslit pairs as the unit and by adjusting the radial distance between the two slits in a unit to realize the arbitrary superposition of two OAM states with different amplitude ratios [40], but exactly varying separations between the two slits was inconvenient, and the replacement of the slit pair with a single slit as the unit for the adjustable amplitude ratio’s superposition would be a novel and interesting method to be developed.

In this paper, we propose a new kind of plasmonic metasurface that combines a chiral spiral with single nanoslits. The proposed structures can manipulate the phases and amplitudes of OAM states continuously and independently at will, which not only enables the superposition of two OAM states with arbitrary TCs but also allows free control of the relative amplitude. We first found that the transmittance coefficients of the converted and transmitted spin components increase and decrease, respectively, with a decrease in the slit width. Using the metasurfaces of nanoslits with adjustable widths, the amplitudes of the two OAM states corresponding to the two spin components can be adjusted, and superposition with different amplitude ratios can be achieved. Essentially, an Archimedean spiral of a variable radius in an azimuthal angle provides a geometry-dependent helical phase; coupled with a rotated nanoslit, a geometric phase of twice the rotated angle is introduced. By combining the chirality and rotation of nanoslit units, we can create and superpose OAM states with arbitrary TCs. Together with the meticulous tuning of the width of the slits to control the amplitudes, the arbitrary superposition of the plasmonic OAM states with continuously changing amplitude ratios is realized. Obviously, the principle and method here are advances over the previous work of superposition of either an equal amplitude with the metasurfaces of a single-slit unit of an identical dimension [39] or a tunable amplitude with those of a slit-pair unit at the adjusted slit distance [40]. We believe that our system provides a powerful platform for plasmonic manipulation and could be utilized in on-chip devices.

## 2. Materials and Methods

Figure 1 shows a schematic diagram of the proposed metasurface for manipulating the superposition of plasmonic OAM states. As shown in Figure 1a, the structures are composed of a spiral-shaped nanoslit that perforates an Au film deposited on a glass substrate. Figure 1b provides a top-view schematic of the metasurface. The radius of the *n*-order spiral can be expressed in polar coordinates as follows: *r_n_* = *r*_0_ + *λ* × mod(*nθ*,2π)/2π, and *r*_0_ represents the minimum radius of each spiral segment. The rotation angle of the slits can be expressed as *β* = *qθ* + *β*_0_ with respect to the *x*-axis, where *q* represents the rotation order of the slit, and *β*_0_ represents the initial angle when the azimuth angle *θ* = 0. When circularly polarized light is incident on the sample from the substrate side, the excited SPP field will contain two terms: one is the original spin component and the other is the converted spin component of opposite handedness with a geometrical phase of twice the rotation angle. As the SPP wave propagates toward the center area of the spiral, the azimuthally varying radius provides a dynamic helical variation phase *nθ*, which is imparted to both the original spin and converted spin components of the SPP to form the superposition of two OAM states near the center point. Further, the transmitting coefficients of two components can be adjusted by the width of each nanoslit, making it possible to perform the superposition of OAMs with different weightings by designing specific widths of the nanoslits. By simultaneously controlling both the phase and amplitude, the superposition of two OAM states *l*_1_ = 3 and *l*_2_ = −3 with arbitrary amplitude ratios is schematically realized, as illustratively shown in Figure 1a.

To demonstrate the properties of wavefield excitation of the nanoslit described above, we carried out numerical investigations on the transmitted wavefields of a single slit at different widths for left circularly polarized light illumination, in which the two spin components are separated and analyzed individually. In the calculation, the nanoslit is set to have a fixed length *L* = 300 nm with the width *w* varying in the range from 175 nm to 300 nm, and the width and length of the slit are parallel to the *x* and *y* axes, respectively. *ρ* and *ξ* denote the normalized coefficients of the original and converted components, respectively. Figure 1c reveals the relationship between *ρ* and *ξ* versus the width of the slit. As shown in the Figure, at the initial width *w* = 175 nm, *ρ = ξ =* 0.5, indicating that the transmitted field contains both original and converted spin components with equal coefficients. With the width changes from 175 nm to 300 nm, the transmitted coefficient *ρ* increases from 0.5 to 1, but the converted coefficient *ξ* decreases to 0. At the maximum slit width *w* = 300 nm, *ρ =* 1 and *ξ* = 0, meaning that the transmitted light field has only an original spin but no conversion component. Especially when the incident polarization is LCP, the transmitted coefficient of the LCP spin component increased from 0.5 to 1 as the width changed from 175 nm to 300 nm, and the coefficient of the RCP spin component decreased from 0.5 to 0. When the illuminating light changed to RCP light, the LCP component coefficient decreased from 0.5 to 0, and the RCP component efficiency increased from 0.5 to 1. Thus, by changing the slit width and the handedness of the incident polarization, the ratio of two spin components can be continuously varied from 0 to 1, enabling superposition with different amplitudes. We also conducted a simulation of the transmittance, reflection, and absorption spectra for the nanoslits with *L* = 300 nm and *w* = 175 nm, 250 nm, and 300 nm, respectively, as shown in Figure 1d. Their transmitted efficiencies for 632.8 nm are 15.7%, 18.2%, and 16.2%, respectively. One of the reasons for the limited efficiency may be the small aspect ratio of the nanoslit for the occupied substrate area.

It is well understood that a nanoslit can be considered an approximate dipole source and excites SPPs along the normal directions (***n***_1_ and ***n***_2_), perpendicular to the two sides of a slit, as shown in Figure 1b. When the nanoslit is illuminated by circularly polarized light, it can be expressed in the polar coordinate system as:(1)$$E_{\mathrm{RCP}/\mathrm{LCP}}=\frac{1}{\sqrt{2}}e^{i\sigma \theta }[e_{r}+i\sigma e_{\theta }]$$
where *σ* = ±1 represents the left circularly polarized (LCP) and right circularly polarized (RCP) lights, and ***e****_r_* and ***e****_θ_* are the radial and azimuthal unit vectors, respectively, the radial projection of the launched SPPs toward the center can form a pattern on the plasmonic field. The corresponding electric field distributed along the radial direction should follow [35]:(2)$$E_{r}(r,\theta )=\frac{1}{\sqrt{2}}[\rho e^{i\sigma \theta }-\xi e^{-i\sigma \theta +i2\sigma \beta }]e_{r}.$$

This demonstrates that the transmitted optical field can be decomposed into two orthogonal spin components: one part preserves the same spin with the incidence, and the other is converted to the reversed spin with an additional phase of 2*σβ*. For the SPP wave, the z-polarization electric field is considered a dominant field component, where *d**E***_spp_ at the field point can be described as [41]:(3)$$dE_{\mathrm{spp}}=e_{z}E_{0Z}e^{-k_{z}z}e^{ik_{r}\cdot (R-r)}E_{r}(r,\theta )rd\theta$$

Obviously, only the converted spin component in the excited SPP field can be modulated by the geometric phase. To obtain the superposition of two vortex states with tunable TCs, a phase factor is necessary to be imposed on the original spin component. Considering that the radial distance from the spiral with geometric charge *n* to the center point is a linear function of the azimuthal angle, it introduces an additional dynamic phase, given that *nθ* contributes to both of the two spin components. Therefore, the phase modulations of the two components can be realized by the variable orientation angles and the changing radial distances of the nanoslits on the segmented spirals. By integrating wavelets from all of the nanoslits and using the representations of Bessel functions, the total SPP field at point Q (*R*, *ϕ*) can be written as:(4)$$\begin{array}{ll} E_{\mathrm{spp}}(R,\phi ) & =e_{z}E_{0Z}e^{-k_{z}z}/\sqrt{2}\int e^{ik_{r}[r_{0}-R\cos(\phi -\theta )]}e^{in\theta }E_{r}(r,\theta )r_{0}d\theta \\ & =e_{z}r_{0}E_{0Z}e^{-k_{z}z}e^{ik_{r}r_{0}}/2\int e^{ik_{r}[r_{0}-R\cos(\phi -\theta )]}\\ & \times \;[\rho e^{i(\sigma +n)\theta }+\zeta e^{i(2\sigma p-\sigma +n)}{}^{\theta +i2\sigma \beta_{0}+i\pi /2}]d\theta \\ & \propto \rho J_{l}{}_{1}\left(k_{r}R\right)e^{il_{1}\phi }+e^{i\varphi }\zeta J_{l}{}_{2}\left(k_{r}R\right)e^{il_{2}\phi }\\ & =\rho |l_{1}>+e^{i\varphi }\xi |l_{2}> \end{array}$$
where |*l*_1_> = *J_l_*_1_(*k_r_R*)exp(*il*_1_*ϕ*) and |*l*_2_> = *J_l_*_2_(*k_r_R*)exp(*il*_2_*ϕ*) represent two OAM states, and *J_l_*_1_(*k_r_R*) and *J_l_*_2_(*k_r_R*) denote the *l*_1_-order and *l*_2_-order Bessel functions of the first kind, respectively. The TCs of the two OAM states are *l*_1_ = *σ* + *n* and *l*_2_ = *σ*(*2q −* 1) + *n*, respectively, and *φ* = 2*σβ*_0_ − π/2 is the phase difference between the two states. The above equation manifests in an output field that is a superposition of two OAM beams with arbitrary TCs and tunable amplitudes. Obviously, the TCs can be flexibly modulated by the incident light *σ*, a rotation order *q* of the nanoslits, and the geometric order *n* of the segmented spiral. Because the amplitude coefficients *ρ* and *ξ* for the OAM beams of the two components are width-dependent, by setting different widths of the nanoslits, a variety of relative amplitude ratios of the OAM states can be achieved.

## 3. Results

To demonstrate the proposed plasmonic metasurface, we performed theoretical calculations and finite-difference time-domain (FDTD) simulations of the wavefields distribution of superposed OAM states. In the simulations, a circularly polarized beam with a wavelength of 632.8 nm is used, and the corresponding SPP wavelength λ = 600 nm. The simulation area is 14 µm × 14 µm, the minimum mesh cell size is 5 nm, and the accuracy is set as two. A perfectly matched layer is used as the boundary conditions’ absorbing boundary to simulate an aperiodic structure to prevent influence from the adjacent periodic cells. The wavefields of the SPPs are mapped by defining the electric field monitors for *E*_z_, and the results are obtained at *z* = 100 nm. The initial radius of the structures *r*_0_ = 4.8 µm, and the end radius *r*_1_ = 6.6 µm. Four types of metasurfaces are designed for generating and manipulating the superpositions of OAM states with different TCs and amplitude ratios. 

With the first type of samples, the equal-weighted superposition of two OAM states with equal but opposite TCs is investigated. The intensity patterns of the transmitted fields by theoretical calculation and FDTD simulation from the metasurfaces are shown in the left panel of Figure 2, respectively. Here, the slit-rotation order *q* is equal to the geometric factor of spiral *n*, the width of the slit *w* = 175 nm, and the initial nanoslit angle *α*_0_ = −π/4; these parameters define the values of the TCs *l*_1_ and *l*_2_ and the phase difference *φ* for the wavefield and realize the superposition of the OAM states *|l>* and *|−l>* with equal amplitudes under the illumination of RCP light. The first column in the left panel in Figure 2 shows the produced by the five metasurfaces, and the magnified views of the central area labeled by white squares are presented in the second column. It can be seen that the results of the FDTD simulation are quite consistent with those of the theoretical calculations given in the third column. The first row depicts the intensity patterns of the first metasurface with the parameters (*q*_1_*, n*_1_*, β*_0_) = (2, 2, −π/4), corresponding to the equal superposition of the OAM states |1> and |−1> under the RCP illumination, which is an interference beam with a two-lobed intensity distribution. For the second metasurface with *q* = *n* = 3, the petal-like intensity images with four lobes are formed as given in the second row. Given the same reason as in the analysis, the four-lobed structured beam is obtained here by the superposition of the OAM states with TCs of two and negative two. Similarly, in the intensity images for the metasurfaces of parameters (*q*, *n*) = (4, 4), (5, 5), and (6, 6), shown in the pictures from the third to the fifth row, respectively, we may see that the superposed states |3> + |−3>, |4> + |−4>, and |5> + |−5> with six, eight, and ten lobes around the center are obtained, and the images demonstrate the consistency of FDTD results with the theoretical predictions that the superposition of the OAM states |*l*> and |−l> with opposite signs will create |2*l*| lobes around the center. Moreover, for this type of metasurface, by designing a metasurface with unequal values of the geometrical order of the spiral *n* and rotational order of slit *q*, the equal-weighted superposition of two OAM states with different TCs (|*l*_1_| ≠ |*l*_2_|) is achieved. To produce such superpositions, we design another five metasurfaces with *n* = 5, but with *q* varying from three to one with increments of 0.5 for two successive metasurface samples. The two OAM states |*l*_1_ = 4> and |*l*_2_ = −(2*q −* 1) + 5> are obtained, respectively, under RCP illumination, when *l*_2_ varied from 0 to 4. The right panel images of Figure 2(e1–e5) show the superposition of OAM states with *l*_2_ = 0, 1, 2, 3, and 4, corresponding to *q* = 3, 2.5, 2, 1.5, and 1, and the OAM state *l*_1_ = 4, respectively, where the first column shows the simulated whole-field distribution, and the second column gives the corresponding magnified views. These simulation results are in good agreement with the theoretical ones shown in the last column. In contrast to the intensity images of the superposed states with opposite signs in the first panel, here, the images of the superposition states with different TCs present more complicated patterns. Specifically, the main lobes of|*l*1 − *l*2| are formed in the intensity images because of the superposition of two OAM states with TCs *l*1 and *l*2. As shown in Figure 2(e1), a central focus point surrounded by four lobes gives a clear demonstration of the existence of the superposed state |0> + |4>. With *l*2 increased from 0 to 3, the lobe number in each image decreased from 3 to 0, demonstrating that the number of lobes of |*l*1 − *l*2|. At the same time, the focus point vanishes and forms a pure vortex with a TC of four as indicated by the phase distribution in the inset of Figure 2(e5).

With the above realization of the equal-weight superposition of the OAM states, the next type of the metasurface is designed to perform the unequal-weighted superposition of two OAM states, in which the amplitudes of the OAM states are arbitrarily adjusted. To this end, each metasurface sample is designed using a specific width of the slits to achieve the specific amplitude coefficients for *ρ* and *ξ* of the two spin components, as analyzed in Figure 1c and expressed by Equation (4). By designing the metasurfaces with nanoslits of different widths, the superposed states *ρ*|*l*> + *ξ*|−*l*> can be realized. For such superposed states, since the two normalized coefficients satisfy |*ρ*|^2^ + |*ξ*|^2^ *=* 1, the evolution of the states with the variation of *ρ* and *ξ* can be described by the Bloch sphere (BS) [37], as shown in Figure 3a, akin to Poincaré sphere for the description of the polarization properties of light. We first design the metasurfaces with parameters (*q*_2_, *n*_2_, *β*_2_) _s1_ = (3, −3, *π*/4), which enables the superposed states *ρ*| 2> + *ξ*| −2> under the illumination of an LCP light, and, here, the subscript s_1_ is for the LCP illumination. The 300 nm length of the nanoslits is the same for all of the metasurfaces, and the widths of the nanoslits *w*_2_ = 300 nm, 250 nm, and 200 nm are used to the design the three samples to generate the superposed states geometrically represented by the three points i, ii, and iii in the northern hemisphere in Figure 3a. The corresponding values of *ρ* and *ξ* can be read in the curves in Figure 1c. The superposed states represented by points iv and v on the southern hemisphere are achieved by the other two metasurfaces with parameters (*q*_2_, *n*_2_, *β*_2_) _s2_ = (3, 3, −π/4) and widths *w* = from 250 nm to 300 nm, respectively; however, the illumination light of the opposite circular polarization, i.e., RCP, gives the interchanged values of *ρ* and *ξ.* Again, the subscript s2 is for the RCP illumination. These results are shown in Figure 3b, where the images from A1 to A5 are the theoretical results, and the small rectangular images under each main image demonstrate the intensities of the two contained vortices *ρ*| *l*> and *ξ*| −l>. The images from B1 to B5 in Figure 3b are the results of FDTD simulations. Here, we note that the metasurfaces with *w* = 175 nm produce the equal-weighted superpositions of the OAM states, corresponding to the point on the equator of the BS, which has been given in the second and the third rows in the left panel in Figure 2. Interestingly, for the metasurface with *w* = 300 nm, the vortex with *l*_2_ = −1 is formed with the intensity pattern shown in Figure 3(A1), which is identical to the case shown in Figure 3(A5), but with the helicity of the phase distribution is reversed. In these two cases, the output field has only the incident spin component, and the change in the handedness of the illuminating circular polarization will change the ratio *ρ*:*ξ* from 1:0 to 0:1. As the width changes in the range from 175 nm to 300 nm, the varied weights of the two vortices will result in a decrease in the azimuthal contrast in the two adjacent lobes, and the intensity profiles appear inhomogeneous. The structured intensity distribution is shown in Figure 3(A3). To further characterize the properties of the superposed OAM states of a high order, we designed metasurfaces with the parameters (*q*_2_, *n*_2_, *β*_2_) _s1_ = (4, −4, *π*/4) for LCP illumination and a metasurface with the parameters (*q*_2_, *n*_2_, *β*_2_) _s2_= (4, 4, −π/4) for RCP illumination, realizing the superpositions of the OAM states *ρ|*3> + *ξ*|−3>. The theoretical and simulated results for these samples with the width of the slits varying from 200 nm to 300 nm are shown in the images C1–C5 and D1–D5, respectively, in Figure 3c.

Next, the third type of metasurface is designed to generate superposed states corresponding to points along another meridian line from the north pole to the south pole on the BS, as shown in Figure 4a, to demonstrate the generation of the superposed states at arbitrary points on the BS. For the illumination by the LCP light, the metasurfaces with parameters (*q*_3_, *n*_3_, *β*_3_) _s1_ = (3, −3, −π/4) enable the generation of the superposed states *ρ*|*l*_1_ = 2> − *ξ*|*l*_2_ = −2>, which can be illustrated as the points on the northern hemisphere. The superposed states in the southern hemisphere are also demonstrated by the metasurface with the parameters (*q*_3_, *n*_3_, *β*_3_) _s2_ = (3, 3, *π*/4) under the illumination of the RCP light. The results are shown in Figure 4b. Compared with the results in Figure 3b, the increment *π/*2 at its initial angle introduces an additional phase difference of *π* between the two superposed eigenstates states, resulting in a *π*/|*l*_1_ − *l*_2_| rotation of intensity profiles. It is clear that the four-lobed structures in Figure 4(A3, B3) are rotated with an angle of 45° in comparison with those in Figure 3(A3, B3). Similarly, the six-lobed structures of the superposed states |3> + |−3> are identical in shape but rotate 30° compared with those in Figure 3. All of the intensity profiles have good consistency with the theoretical results, verifying the highly efficient generation of arbitrary superposed states with the designed metasurfaces.

The proposed scheme can also be extended to manipulate the hybrid superposition of two OAMs with different TCs. The fourth type of metasurface is designed to generate the hybrid superposition of two OAM states with different TCs and unequal weights. The geometric parameters are (*q*_4_, *n*_4_, *β*_4_) _s1_ = (4, 3, −π*/*4) and (*q*_4_, *n*_4_, *β*_4_) _s2_ = (4, −5, *π/*4). The superposition states *ρ|*2> + *ξ*|−4> and *ρ|*−4> + *ξ*|2> are observed for RCP and LCP illuminations, respectively. As the width of the slit is changed from 300 nm to 200 nm with the incidence spin reversed from *σ* = −1 to *σ* = +1, the intensity profile of the superposed states starts from the vortex stated of |2> with the smaller dark hole as shown in Figure 5(a1), presenting the OAM state with *l*_1_ = 2 for the superposed state of *ρ|*2> + *ξ*|−4> at *ρ* = 1 and *ξ* = 0, and it evolves to that of the final vortex states of |−4> with the larger dark hole as in Figure 5(a5), as the result of *ρ|*−4> + *ξ*|2> at *ξ* = 0. For the transition of the intermediate states of the hybrid superposition, the six lobes of the intensity profiles begin to appear and become highly contrasted with the coefficient ratio of the *ρ* and *ξ* changes from the extrema 1:0 to the equal-weight proportion 1:1, as demonstrated in Figure 5(a2–a4). The simulated intensity profiles are in remarkable agreement with the theoretical calculations. This agreement confirms that the proposed metasurface performs well for generating an arbitrary superposition of OAM states with different TCs and tunable amplitude ratios.

Though our work only focuses on the theoretical principle and design method for metasurfaces, they are validated by the results of the FDTD simulations and theoretical calculations; here, we give further analysis of the possible experimental measurements and demonstrations that would not be too difficult to be realized [40]. Regarding the experimental performance, the designed metasurface samples can be directly fabricated by focusing on ion-beam etching (FIB) by directly using the nanostructures of the sample obtained from the results of FDTD simulations. The fabricated sample can be measured in an optical setup similar to that described in Reference [40]. A near-field scanning optical microscope (SNOM) in collection mode can be used to measure the intensity profiles of the SPP field. A 632.8 nm wavelength laser source passes through a linear polarizer and a quarter-waveplate to generate circularly polarized incident light. The laser beam would be focused by an object lens (20×, numerical aperture: 0.45) onto the samples from the substrate side. A probe with an aluminum-coated fiber tip could be used to collect information on the optical field and mapped on a computer. Thus, measurements of the generated superposed OAM states and the verifications for the feasibility of the metasurface design may be realized.

## 4. Discussions

The superposition of OAM states in optical beams has been studied extensively, as achieved and demonstrated in our previous works, Ref. [40] and Ref. [39]. In the work of Ref. [40], the nanoslit pair containing two perpendicular nanoslits is used as the unit in the nanostructure design to convert the incident light into two orthogonal spin components. The amplitudes of these two spin components can be modulated by the phase difference originating from tuning the radial distance of two slits, which enables the differently weighted superposition of two OAM states corresponding to the two spin components. In addition, the plasmonic metasurface with the single-sized nanoslit used as the unit can only generate the equal-weighted superposition of OAM states and lacks the manipulation for the amplitudes of the two OAM states in Ref. [39]. In contrast to the studies in the literature, the present work makes use of nanoslits with diverse dimensions as a unit for constructing the metasurface. The functionality of this kind of nanoslit unit is that it can be used to implement amplitude modulation of two OAM states by changing the width of each slit, thereby achieving the amplitude-tunable superposition of plasmonic OAM states. Thus, compared with the previous studies, our present work provides a totally different and remarkably innovative method for the manipulation of plasmonic OAM modes. Moreover, our FDTD simulations show that the designed metasurfaces possess broadband characteristics; the OAM superposed states can be obtained in a wavelength ranging from 560 nm to 670 nm by adjusting the radial distance between the two adjacent spirals. Take the metasurface with the width of the slit *W* = 200 nm, for example; the proportion on the incident field converted to SPPs in the designed metasurface, defined as the ratio of the wavefield energy to the light energy incident on the area of the metasurface, is 46.5%. The conversion efficiency uses the ratio of wavefield energy propagating to the center area to form a pattern of the OAM superposition to the light energy incident on the area of the nanoslit, and the conversion efficiency is obtained as 20.8%. The low efficiency mainly results from the propagating losses from the edge of the metasurface to the center area. The metasurface efficiency can be increased by reducing the metasurface diameter or by choosing highly efficient elements.

## 5. Conclusions

In summary, we propose a strategy for designing a plasmonic metasurface to realize the generation and superposition of OAM states in the SPPs field. The proposed metasurface combines both the geometric phase and dynamic phase by controlling the orientation angle of each nanoslit unit and the geometric order of the spiral, leading to the superposition of two OAM states with different TCs to be achieved in a flexible manner. In addition, the relative amplitude of two OAM states also can be independently modulated by tuning the width of the nanoslits. We designed four types of metasurfaces for conducting the equal- and unequal-weighted superpositions and the hybrid superposition of OAM states with different TCs and various amplitude ratios, theoretically and numerically. The numerically calculated results show a good agreement with the theoretical models. Our results provide a new approach for flexible manipulation of the wavefronts of SPPs, which has potential applications in communication, information processing, and high-performance plasmonic devices.

## Figures and Tables

**Figure 1 materials-15-06334-f001:**
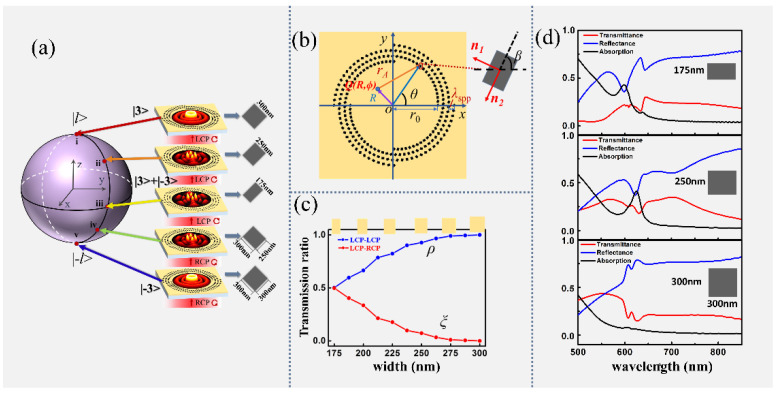
(**a**) Configuration of metasurface transforming an incident circularly polarized light into superposed OAM states corresponding to the points of the Bloch sphere. (**b**) Diagram of a metasurface pattern with parameters (*q*, *n*, *β*) = (4, 4, π/4); inset, enlarged view of one nanoslit. (**c**) Transmitted coefficients of original and converted spin components vary with the width of the nanoslit. (**d**) The transmittance, reflection, and absorption spectra for the slits with widths *w* = 175 nm, 250 nm, and 300 nm, respectively.

**Figure 2 materials-15-06334-f002:**
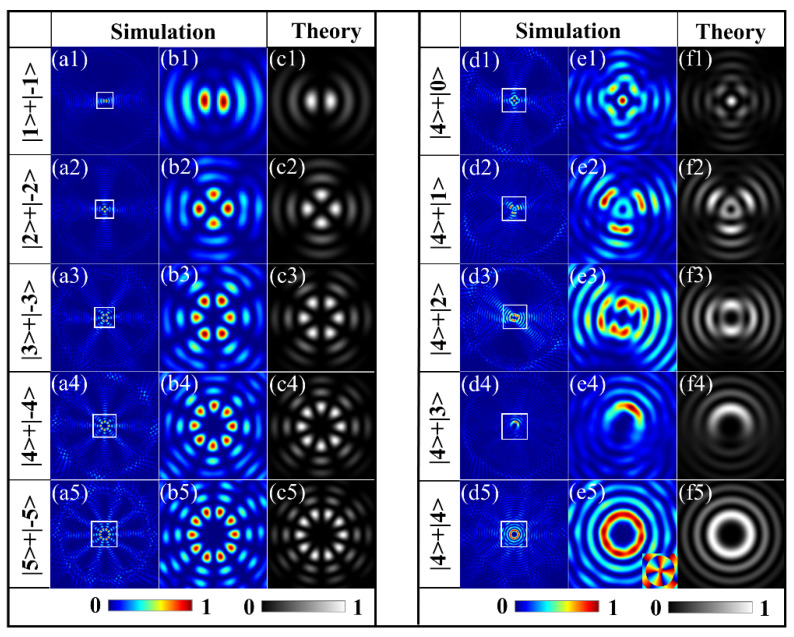
Simulated and theoretical intensity profiles of the equal-weighted superposition of two OAM states with different TCs. Left panel: equal-weighted superposition of OAM states with the same TCs but opposite signs |*l*> + |−*l*>, (**a1**–**a5**) the whole-field intensity distributions, (**b1**–**b5**) the corresponding magnified view of the center area labeled by the white square in the first column, (**c1**–**c5**) theoretical results. The right panel (columns (**d1**–**d5**), (**e1**–**e5**), and (**f1**–**f5**) shows the same results as the left panel but for the superposed OAM states with different TCs. The inset is the corresponding phase distribution.

**Figure 3 materials-15-06334-f003:**
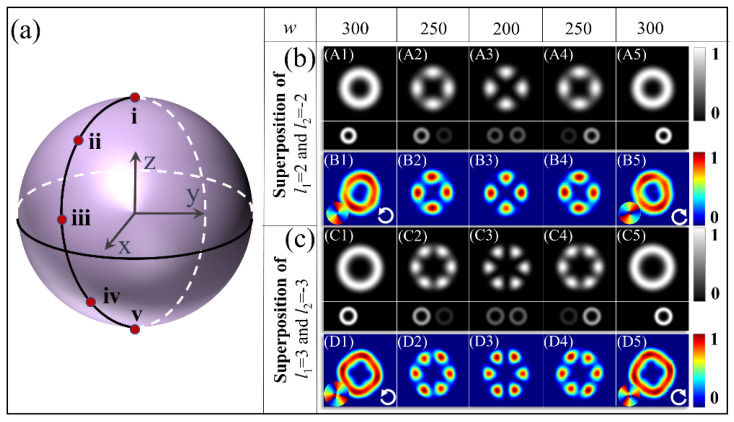
Theoretical and simulated intensity profiles of the superposition of two OAM states with various amplitude ratios. (**a**) BS for representing the superposed states. (**b**) The results for superposed states of *ρ*|2> + *ξ*|−2>: (**A1**–**A5**) Theoretical results with the insets showing the two superposed eigenstates. (**B1**–**B5**) The corresponding FDTD-simulated results of the area near the center point of the structure. (**c**) The results for superposed states of *ρ*|3> + *ξ*|−3>: (**C1**–**D5**) The calculated theoretical and corresponding simulated intensity profiles. The resets in the lower left corner of (**B1**), (**B5**,**D1**,**D5**) present the phase distributions.

**Figure 4 materials-15-06334-f004:**
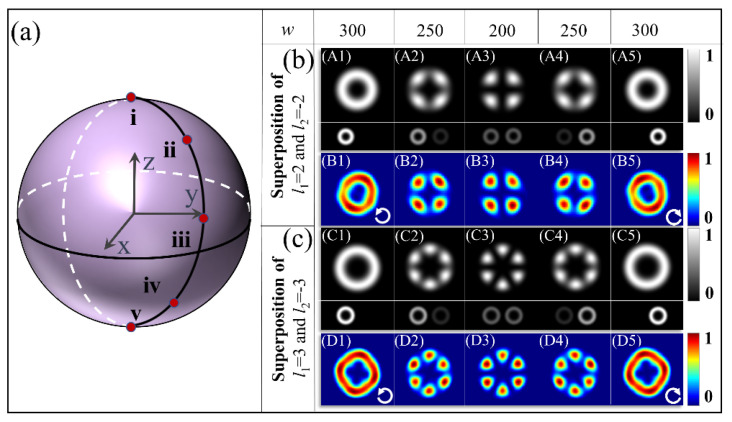
Theoretical and simulated intensity profiles of the superposition of two OAM states with various amplitude ratios. (**a**) BS for representing the superposed states. (**b**) The results for superposed states of *ρ*|2> − *ξ*|−2>: (**A1**–**A5**) Theoretical results with insets showing the two superposed eigenstates. (**B1**–**B5**) The corresponding FDTD-simulated results of the area near the center point of the structure. (**c**) The results for superposed states of *ρ*|3> − *ξ*|−3>: (**C1**–**D5**) The calculated theoretical and corresponding simulated intensity profiles.

**Figure 5 materials-15-06334-f005:**
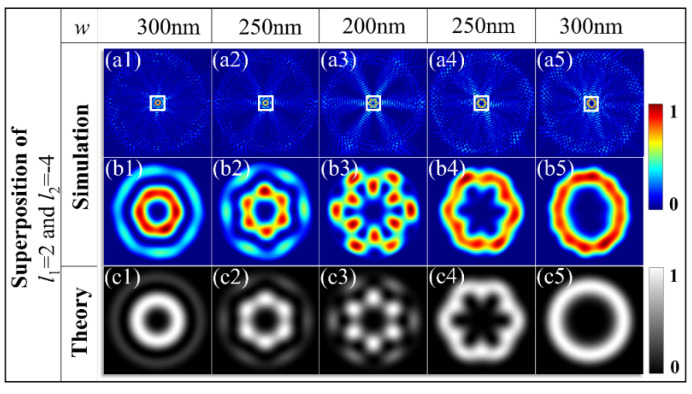
Theoretical and simulated intensity profiles of the hybrid superposed state *ρ|*3> + *ξ*|−1>; (**a1**–**a5**) the whole-field intensity distributions, (**b1**–**b5**) the corresponding magnified view of the center area labeled by the white square in the first column, (**c1**–**c5**) theoretical results.

## Data Availability

Not applicable.

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
