# Peer review of "Metasurfaces for Amplitude-Tunable Superposition of Plasmonic Orbital Angular Momentum States"

_materials, 2022, doi:10.3390/ma15186334_

Round 1

Reviewer 1 Report

The paper is very well written and presented. The topic of “superposition” is also a very current research field. The concepts, presented, are demonstrated by numerical modeling.

The title of the article nicely summarizes what research issue the paper is addressing: “Metasurfaces for amplitude-tunable superposition of Plasmonic orbital angular momentum states.

The concept is not at all original. It is not filling any existing gap either. Other people, including this group, have already published similar concept of superposition of plasmonic orbital angular momentum states, as has been noted by the authors by citing others and their own publications, Ref.#38.

This paper adds a bit more detailed presentation compared to this group’s previous publication on “nano structure” cited as Ref.# 38; compared to the current manuscript on “metasurfaces”.

Authors use the word “demonstration”, which implies, to me, real experimental demonstration, not just simulation with beautiful and colorful computer graphics. Since the authors have already published the core concept [Ref.#38], may be, they should consider re-submitting the paper with some real experimental demonstrations.

They posed their research issue as “….superposition of plasmonic orbital angular momentum states”. Their theory and simulations are consistent with what have posed in the abstract.

References are relevant and cited at proper contexts within the paper.

I have no specific comments on their simulated computer graphics, which are presented very nicely. The paper is also written very well.

Reviewer 2 Report

The authors present interesting an interesting perspective for the realization of OAM states with multiple topological charges using patches using spiral nanoslits. The following observations must be addressed.

  1. Little to no information on the FDTD simulation parameters has been offered (boundary conditions, meshing, accuracy) - here, the authors need to introduce details on the simulation conditions.
  2. No spectroscopic response assessment is performed. It is useful to know the fraction of the field that is transmitted through or reflected by the surface, in order to qualify the efficiency - here, the authors have to introduce some reflection,transmission and absorption graphs detailing the spectral response of the metasurface

3.     The introduction can be further improved by adding recent relevant studies with ring-rod structures: 10.1016/j.jqsrt.2020.107209 and 10.3390/polym13111860.

I can recommend publication after a Major Revision in which the observations have been addressed.

Reviewer 3 Report

The authors present theoretical and corroborating simulation results for a plasmonic metasurface giving rise to the generation and superposition of orbital angular momentum states using surface plasmon polariton excitations.

When circularly polarized light is incident on the structure, the excited SPPs are a superposition of the original field spin component, and the converted spin component of opposite handedness, with the geometrical phase twice the rotation angle.And as an SPP wave propagates toward the center area of the proposed spiral structure, the azimuthally varying radius provides a dynamical helical variation phase, imparted to both the original spin and the converted spin components of the SPP, forming the superposition of two OAM states near the center point.

The manuscript is clearly written, and the corroborated results appear technically sound. This is an interesting work, potentially to scientists working on metasurfaces and/or OAM. It could be published on the journal after the authors address the following points:

1. What is the proportion on incident field converted to SPPs in this structure? can this efficiency be increased?

2. Why is 'ksi' only up to 0.5? can this efficiency be increased further?

3. Over how large bandwidth can this structure operate?

4. What is the role of realistic levels of losses in this structure? how are the above efficiencies affected?

5. Important works on SPPs in plasmonic / metamaterial structures, such as e.g. Physical Review B 72, 075405 (2005), Physical Review B 73, 085104 (2006), etc, are currently missing and should be cited in a revised version of the work.

Round 2

Reviewer 2 Report

Please correct the author name on reference 34 from Dnil to Danila (just like in ref 35). Other than that, the article can be Accepted for publication.

Reviewer 3 Report

I have read the revised manuscript, and the authors' replies to my previous comments. The authors have done a good job in nicely revising the work, and have indeed satisfactorily addressed all raised questions. As such, from my perspective, this work is now ready to appear in the journal as is.